# Association of Left Atrial Size Measured by Non-Contrast Computed Tomography with Cardiovascular Risk Factors—The Danish Cardiovascular Screening Trial (DANCAVAS)

**DOI:** 10.3390/diagnostics12020244

**Published:** 2022-01-19

**Authors:** Maise Høigaard Fredgart, Jes Sanddal Lindholt, Axel Brandes, Flemming Hald Steffensen, Lars Frost, Jess Lambrechtsen, Marek Karon, Martin Busk, Grazina Urbonaviciene, Kenneth Egstrup, Lida Khurrami, Oke Gerke, Axel Cosmus Pyndt Diederichsen

**Affiliations:** 1Department of Cardiology, Odense University Hospital, 5000 Odense, Denmark; Maise.Hoigaard.Fredgart@rsyd.dk (M.H.F.); axel.brandes@rsyd.dk (A.B.); Lida.Khurrami2@rsyd.dk (L.K.); 2Odense Patient Data Explorative Network (OPEN), Odense University Hospital, 5000 Odense, Denmark; 3Department of Cardiothoracic and Vascular Surgery, Odense University Hospital, 5000 Odense, Denmark; Jes.Sanddal.Lindholt@rsyd.dk; 4Cardiovascular Centre of Excellence (CAVAC), Odense University Hospital, 5000 Odense, Denmark; 5Elitary Research Centre of Individualised Medicine in Arterial Disease, Odense University Hospital, 5000 Odense, Denmark; 6Department of Cardiology, Hospital Lillebælt, 7100 Vejle, Denmark; Flemming.Hald@rsyd.dk (F.H.S.); Martin.Busk@rsyd.dk (M.B.); 7Department of Cardiology, Regional Hospital Central Jutland, 8600 Silkeborg, Denmark; larfrost@rm.dk (L.F.); grazina.urbonaviciene@viborg.rm.dk (G.U.); 8Department of Cardiology, Svendborg Hospital, 5700 Svendborg, Denmark; jess.lambrechtsen@rsyd.dk (J.L.); Kenneth.Egstrup@rsyd.dk (K.E.); 9Department of Medicine, Nykøbing Falster Hospital, 4800 Nykøbing Falster, Denmark; mkar@regionsjaelland.dk; 10Department of Nuclear Medicine, Odense University Hospital, 5000 Odense, Denmark; oke.gerke@rsyd.dk

**Keywords:** left atrial size, non-contrast computed tomography, echocardiography, cardiovascular risk factors

## Abstract

Left atrium (LA) size is associated with adverse cardiovascular events. The purpose of this study was to investigate the association of LA enlargement measured by non-contrast CT (NCCT) with traditional cardiovascular risk factors. Individuals aged 60–75 years from the population-based multicentre Danish Cardiovascular Screening (DANCAVAS) trial were included in this cross-sectional study. The LA was manually traced on the NCCT scans, and the largest cross-section area was indexed to body surface area. All traditional risk factors were recorded, and a subgroup received an echocardiographic examination. We enrolled 14,987 individuals. Participants with known cardiovascular disease or lacking measurements of LA size or body surface area were excluded, resulting in 10,902 men for the main analysis and 616 women for a sensitivity analysis. Adjusted multivariable analysis showed a significantly increased indexed LA size by increasing age and pulse pressure, while smoking, HbA1c, and total cholesterol were associated with decreased indexed LA size. The findings were confirmed in a supplementary analysis including left ventricle ejection fraction and mass. In this population-based cohort of elderly men, an association was found between age and pulse pressure and increasing LA size. Surprisingly, smoking, HbA1c, and total cholesterol were associated with a decrease in LA size. This indicates that the pathophysiology behind atrial cardiomyopathy is not only reflected by enlargement, but also shrinking.

## 1. Introduction

Left atrial (LA) size is associated with adverse cardiovascular events and has been recognised to be an independent predictor of cardiovascular outcome [1,2,3,4]. As an increased LA size is connected to the risk of atrial fibrillation and heart failure, the risk factors causing LA dilation would therefore be expected to be similar to those leading to atrial fibrillation and heart failure. Hypertension and obesity are some of the most important modifiable risk factors for atrial fibrillation; in addition, diabetes, obstructive sleep apnoea, smoking and alcohol use are also known to increase the risk of atrial fibrillation [5]. However, atrial cardiomyopathy is not only reflected by dilatation, but also fibrosis, contractile dysfunction, and arrhythmias. Thus, examining associations between LA size and risk factors could improve our understanding of mechanisms behind atrial cardiomyopathy.

An increasing number of patients are examined by cardiac non-contrast computed tomography (NCCT) to evaluate coronary artery calcium (CAC) score, as this is the most important risk marker of cardiovascular disease [6,7]. We recently performed approximately 15,000 NCCT scans in the population-based Danish Cardiovascular Screening (DANCAVAS) trial to evaluate the benefit of a cardiovascular screening [8]. As these NCCT scans are appropriate for measurements of LA size, we aimed with this study to evaluate associations between the risk factors and LA size in the DANCAVAS population.

## 2. Materials and Methods

### 2.1. Study Population

Participants in this cross-sectional study were recruited from the DANCAVAS trial [8]. The study began in September 2014. Both men and woman were invited in the beginning, but as a pilot study concluded that women were not likely to benefit cost-effectively, only men were recruited from May 2015 [9]. All men aged 60–75 years living on the Island of Funen and the surrounding communities of Vejle, Silkeborg and Nykøbing Falster were identified and randomly invited for the cardiovascular screening program. The screening program included a low-dose NCCT scan, among other things. In this substudy, we excluded participants without an LA size measurement, missing body surface area (BSA), and/or with known cardiovascular disease (prior myocardial infarction, coronary revascularisation, stroke, atrial fibrillation and surgical intervention for valvular heart disease, aortic aneurysm or peripheral arterial disease).

Study data were collected and managed using REDCap data management, hosted at the Odense Patient data Explorative Network. The study was registered at http://www.isrctn.com/ (ISRCTN12157806).

### 2.2. Non-Contrast Cardiac Computed Tomography

Various CT scanners were used. Two centres used a Siemens Flash (Gantry rotation time 0.28 s, 3.0 mm collimation, acquisition 128 × 0.6 mm, 120 kV tube voltage, 90 mAs tube current) or Siemens Force (Gantry rotation time 0.25 s, 3.0 mm collimation, acquisition 38 × 1.2 mm, 120 kV tube voltage, 80 mAs tube current) (Siemens Healthcare Solutions, Erlangen, Germany). Imaging was prospectively ECG-triggered at 70% of the R-R interval if the heart rate was <75 bpm or at 250 ms after the QRS-complex if heart rate was >75 bpm for the Siemens Flash, and at 70% of the R-R interval if heart rate was <75 bpm or at 300 ms after the QRS-complex if heart rate was >75 bpm for the Siemens Force. A third centre used a Philips iCT 256 slice scanner (Gantry rotation time 0.27 s, 2.5 mm collimation, auto acquisition 128 × 0.625, 112 × 0.625 or 96 × 0.625 mm, 120 kV tube voltage, 50 mAs tube current) (Phillips, Amsterdam, Holland). Imaging was prospectively ECG-triggered at 75% of the R-R interval at all heart rates. A fourth centre used a Toshiba Aquilion One 320 slice scanner (Gantry rotation time 0.35 s, acquisition collimation 320 × 0.5 mm, 120 kV tube voltage, 28 mAs tube current) (Toshiba, Tokyo, Japan). Imaging was prospectively ECG-triggered at 75%, exposure window 450 ms, if the heart rate was <65 bpm or at 40%, exposure window 450 ms, if heart rate was >65 bpm. The fifth centre used a GE Healthcare Revolution scanner (Gantry rotation time 0.28 s, 2.5 mm collimation, Smart coverage acquisition 256 × 0.625, 224 × 0.625 or 192 × 0.625 mm, 120 kV tube voltage, 15 mAs tube current) (GE, Chicago, IL, USA). Imaging was prospectively ECG-triggered at 75% of the R-R interval if heart rate was <75 bpm or at 350 ms after the QRS-complex if heart rate were >75 bpm.

### 2.3. LA Size Measurement

Seven trained radiographers, blinded to all clinical data, measured LA size on a SyngoVia (Siemens Healthcare Solutions, Erlangen, Germany) workstation. LA area was manually traced in axial slices at the level of the mitral annulus excluding the pulmonary veins, and the largest traced cross-section area was chosen (Figure 1). LA area was indexed for BSA, thereby calculating LA area index (LAindex = LAarea/BSA). To assess inter- and intra-observer variability, LA area measurements were repeated in 20 participants by each of the seven radiographers and in 140 participants by a second reader (M.H.F).

### 2.4. Risk Factors

BSA was calculated using Du Bois formula [10]. Known hypertension was defined as use of antihypertensive medication, or a systolic blood pressure > 160 mmHg or diastolic blood pressure > 100 mmHg, measured during the ankle-brachial index measurement, as described in DANCAVAS [9,11]. Diabetes was defined as current antidiabetic treatment, self-reported diabetes, or HbA1c > 48 mmol/mol. Standardized methods were used to determine high- and low-density lipoprotein (HDL and LDL) and total cholesterol. Seven radiographers measured CAC score and valve calcification using the Agatston method [12].

### 2.5. Echocardiography

A supplementary echocardiography was performed in a subset of participants [13] according to recommendations from the European Echocardiographic Society [14]. All images were stored, and offline analyses were subsequently performed blinded for clinical data. The measurements included in this publication are left ventricle ejection fraction (EF) using Simpson’s biplane method, and left ventricle mass derived from linear 2D.

### 2.6. Statistical Analysis

In all analyses, LA area was indexed to BSA. Continuous variables are presented as mean ± standard deviation or 95% confidence interval (CI) and categorical data as numeric (percentage). Non-normally distributed continuous variables are shown as median and interquartile range. Assumptions of normal distributions in data were assessed visually using quantile–quantile plots and numerically by means of the skewness and kurtosis tests for normality proposed by D’Agostino, Belanger, and D’Agostino [15,16]. Scatterplots of indexed LA area and explanatory variables were used to investigate functional relationships possibly suggesting normalising transformations for some variables. Univariate and multivariable linear regression using robust variance estimation were performed to assess the association between cardiovascular risk factors and LA area index. All variables were included in the multivariable model except for collinear variables such as diastolic and systolic blood pressure; here, the most clinically significant variable was chosen. To assess inter- and intra-observational differences Bland–Altman limits of agreement were performed with exact 95% CI for the limits of agreement [17,18] and supplemented by intraclass and Pearson’s correlation coefficients. A *p*-value of <0.05 was considered significant for all other analyses. STATA version 16.0 (StataCorp LP, College Station, TX, USA) was used for statistical analysis.

## 3. Results

A total of 14,987 individuals participated in the screening trials. Of these, 3469 met the exclusion criteria, leaving 10,902 men and 616 women in this study (Figure 2).

Mean age was 67 versus 68 years for men and women, respectively (Table 1). LA size was larger in men (22.7 cm^2^; 95% CI 22.6–22.8) compared to women (19.4 cm^2^; 95% CI 19.1–19.7), but there was no difference between men (11.1 cm^2^/m^2^ (95% CI 11.0–11.1) and women (10.9 cm^2^/m^2^ (95% CI 10.7–11.1) when adjusting LA size for BSA. Risk factors differed among sexes: more men had diabetes, were current or former smokers, and had higher BMI, while more women had hypertension and dyslipidaemia.

In the main analysis only including men, increasing age was associated with increasing LA size. In the multivariable analysis, LA size increased 0.06 cm^2^/m^2^ (95% CI 0.05–0.07, *p* < 0.001) per one-year increase in age, Table 2.

There was a negative association between smoking and LA size in the multivariable analysis. Former smokers had a decreased LA size of 0.42 cm^2^/m^2^, while LA size decreased by 1.10 cm^2^/m^2^ in current smokers, Table 2.

Hypertension including systolic blood pressure, diastolic blood pressure, pulse pressure and anti-hypertensive treatment were significantly associated with an increased indexed LA size in univariate analysis. In the multivariable analysis, pulse pressure was chosen as the clinically most relevant variable, and the association between pulse pressure and increased LA size remained significant (0.03 cm^2^/m^2^ per mmHg), Table 2. Exclusion of participants in anti-hypertensive treatment did not cause any changes (0.03 cm^2^/m^2^ per mmHg), Table 3.

HbA1c and diabetes were associated with decreased LA size, and this association persisted in the multivariable analysis as HbA1c was associated with a decreased LA size (0.02 cm^2^/m^2^ per mmol/mol). In participants without diabetes, HbA1c remained associated with decreased LA size (0.01 cm^2^/m^2^ per mmol/mol).

**Table 3 diagnostics-12-00244-t003:** Linear regression of LA area index (cm^2^/m^2^) in men—a subgroup analysis.

	Univariate			Multivariable ^#^		
Variable	Δ LA Area Index (95% CI)	*p*-Value	R-Squared	Δ LA area index (95% CI)	*p*-Value	R-Squared
No anti-hypertensive treatment (*n* = 6693)						
Pulse blood pressure (mmHg) *	0.031 (0.027; 0.035)	<0.0001	0.036	0.028 (0.024; 0.032)	<0.001	0.070
No anti-diabetic treatment (*n* = 9976)						
HbA1c (mmol/mol) *	−0.009 (−0.018; −0.001)	0.04	0.0004	−0.010 (−0.018; −0.001)	0.03	0.067
No lipid-lowering treatment (*n* = 8319)						
Total cholesterol (mmol/L) *	−0.11 (−0.17; −0.06)	<0.0001	0.0020	−0.13 (−0.19; −0.08)	<0.001	0.072

* By 1 unit increment. ^#^ Adjusted for age, smoking, pulse blood pressure, HbA1c, total cholesterol.

Total cholesterol and LDL were associated with decreased LA size, while HDL was associated with increased LA size. In the multivariable analysis, a decrease of 0.11 cm^2^/m^2^ in LA size per mmol/L increase in total cholesterol was observed. These findings persisted in a subgroup analysis of participants without lipid-lowering treatment (0.13 cm^2^/m^2^ per mmol/L). As the CT scan acquisition depended on the patient heart rate, some scans were performed in the systole (patients with high heart rates) and some in the diastole (patients with low heart rates). As atrial volume is maximal in systole and minimal in diastole, this causes a possible bias. Appendix A shows the respective analyses stratified by heart rate. We found no substantial differences in subgroups of participants with heart rates below 65 vs. heart rates 65 to 75 vs. heart rates above 75.

An echocardiography was available in 757 of the 10,902 men. After inclusion of left ventricle EF and mass, age and pulse pressure remained positively associated with LA size, while smoking, HbA1c and total cholesterol remained negatively associated with LA size, Table 4. In this supplementary model, increasing left ventricle EF was associated with decreased LA size, while increasing left ventricle mass was associated with increased LA size. If the echocardiographic measurement of left atrial volume was used instead of the CT-based measurement, current smoking kept associated with decreased left atrial volume, and left ventricular mass was associated with increased left atrial volume, Table 4. The remaining variables lost significance, but all retained the direction (non-significantly associated with a decreased or increased left atrial volume). The diastolic measurement E/e was not associated with LA size, and was accordingly excluded from the model. Additionally, CAC score, AVC score, and ascending aortic diameter were all associated with increased indexed LA size, Table 2. As neither CAC nor AVC score are risk factors to LA size, they were not included in the multivariable model.

In the univariate analysis including women, age was positively associated with LA size (0.07 cm^2^/m^2^ (95% CI 0.01–0.12, *p* = 0.01)), but lost significance in the multivariable analysis, Table 5. Pulse pressure was associated with increasing LA size (0.03 cm^2^/m^2^ per mmHg) in the multivariable analysis, while total cholesterol was associated with decreasing LA size (0.22 cm^2^/m^2^ per mmol/L). In current smokers, LA size decreased, with 0.77 cm^2^/m^2^ (95% CI −1.33; −0.21, *p* = 0.007) in the univariate analysis, but lost significance in the multivariable analysis. Former smoking and HbA1c was not associated with LA size in the univariate or multivariable analysis.

**Table 4 diagnostics-12-00244-t004:** Linear multivariable regression of left atrium area (cm^2^/m^2^) by cardiac CT (*n* = 757) or volume (mL/m^2^) derived by echocardiographic (*n* = 709)—a subgroup analysis including echocardiographic measurements.

	Atrial Area (cm^2^/m^2^) Derived by Cardiac CT		Atrial Volume (mL/m^2^) Derived by Echocardiographic	
Variable	Δ LA Area Index (95% CI)	*p*-Value	Δ LA Area Index (95% CI)	*p*-Value
Age (years) *	0.08 (0.02; 0.14)	0.01	0.05 (−0.15; 0.25)	0.61
Smoking				
Former	−0.93 (−1.31; −0.55)	<0.001	−0.98 (−2.19; 0.24)	0.12
Current	−1.30 (−1.80; −0.80)	<0.001	−2.50 (−4.10; −0.91)	0.002
Pulse pressure (mmHg) *	0.02 (0.01; −0.03)	<0.01	0.03 (−0.01; 0.07)	0.10
HbA1c (mmol/mol) *	−0.03 (-0.05; −0.01)	<0.01	−0.06 (−0.13; 0.00)	0.06
Total cholesterol (mmol/L) *	−0.17 (-0.32; −0.01)	0.04	−0.27 (−0.77; 0.22)	0.28
Left ventricle ejection fraction (%) *	−0.03 (−0.06; −0.00)	0.04	−0.08 (−0.17; 0.00)	0.054
Left ventricle mass (g) *	0.01 (0.01; 0.01)	<0.001	0.04 (0.03; 0.05)	<0.001

* By 1 unit increment.

**Table 5 diagnostics-12-00244-t005:** Linear regression of left atrium area index (cm^2^/m^2^) in women (*n* = 606).

	Univariate			Multivariable (R-Squared = 0.082)	
Variable	Δ LA Area Index (95% CI)	*p*-Value	R-Squared	Δ LA Area Index (95% CI)	*p*-Value
Age (years) *	0.07 (0.01; 0.12)	0.01	0.0097	0.02 (−0.03; 0.08)	0.42
Smoking					
Former	−0.09 (−0.48; 0.30)	0.64	0.0117	−0.10 (−0.49; 0.29)	0.62
Current	−0.77 (−1.33; −0.21)	0.007		−0.54 (−1.10; 0.03)	0.06
Pulse pressure (mmHg) *	0.04 (0.02; 0.05)	<0.0001	0.0641	0.03 (0.02; −0.05)	<0.001
HbA1c (mmol/mol) *	0.03 (−0.01; 0.07)	0.11	0.0044	0.01 (−0.03; 0.05)	0.57
Total cholesterol (mmol/L) *	−0.20 (−0.77; −0.03)	0.02	0.0083	−0.22 (−0.40; −0.05)	0.01

* By 1 unit increment.

Inter- and intra-observer agreement of LA measurements were performed in 140 participants with a Pearson’s correlation of r = 0.98 and r = 0.97 (*p* < 0.0001). Agreement by Bland–Altman plots are shown in Figure 3 with a mean difference in LA area of −0.02 cm^2^ and 0.58 cm^2^, the limits of agreement were −2.36 to 2.33 and −2.44 to 3.55, and outer 95% confidence limits of these were −2.68 and 2.65, as well as –3.03 and 4.19 in inter- and intra-observer analyses, respectively. Intraclass correlation coefficients (with respective 95% confidence intervals) were 0.98 (0.97–0.99) for the inter- and 0.96 (0.95–0.97) for the intra-observer comparison.

## 4. Discussion

We examined the association between indexed LA area measured by NCCT and classical cardiovascular risk factors in participants without any overt cardiovascular disease. Age and hypertension were associated with LA size, but surprisingly we experienced that not only smoking, but also diabetes and dyslipidaemia were associated with decreased LA size.

Our data demonstrate that a yearly increase in age is associated with an increased LA area. As both atrial size and risk of AF increase by age, remodelling and atrial dysfunction are seen as a direct consequence of natural aging; however, studies using echocardiographic data found no increase in maximal LA volume with normal aging [19]. Atrial enlargement may therefore be due to accumulated risk factors and ventricular pathology seen with increasing age rather than caused by the aging process itself. When adjusting our analysis for all other cardiovascular risk factors, a somewhat smaller effect on atrial area was seen; however, age remained independently associated with LA size. As expected, hypertension, pulse pressure and systolic and diastolic blood pressure were associated with an increased LA area, and the LA area remained significantly increased when the analysis was adjusted for other cardiovascular risk factors. The left ventricle and conditions herein have a direct effect on the LA. As hypertension affects both ventricular filling pressure and ventricular diastolic dysfunction, this will inevitably lead to an increased atrial pressure causing increased wall tension and hence facilitate remodelling. Furthermore, an independent association of LA volume with diastolic dysfunction has been demonstrated [20]. Adjusting our analysis for left ventricle EF and mass did not alter the findings concerning the association between pulse pressure and LA area.

Among classic risk factors, smoking is a key contributor to cardiovascular disease. Several studies have found smoking to be associated with incident AF, and meta-analyses of prospective studies showed a dose-dependent association [21,22]. Tobacco use has several adverse effects and leads to conditions such as diabetes [23] and hypertension [24,25,26], which are known risk factors for AF. Nicotine has been associated with atrial fibrosis, which impairs intracellular conduction leading to arrhythmias [27]. Nicotine has also been shown to be a potent blocker of ion channels in the atrial myocytes [28]. In our study, a negative correlation between both current and former smoking and LA area was seen. As smoking is associated with AF and an indisputable relationship exists between LA size and AF, our findings seem paradoxical. The pathophysiological mechanisms behind these findings are not fully clarified. We examined individuals without know cardiovascular disease, leaving out participants with known AF. One could therefore speculate that our results illustrates the pathophysiological changes of smoking preceding disease manifestation and atrial enlargement occurs later in the development of AF. However, this cannot fully explain our findings. As others have described a correlation of decreased LA size with smoking [29], our results are not likely to be coincidental. Hyper-inflated lungs, as seen in smokers with emphysema, could have decreased intrathoracic blood volume and could thereby decrease preload, leading to decreased filling and dimensions of the chambers in the heart [30,31].

A similar paradoxical correlation was found for diabetes and dyslipidaemia. The negative association between HbA1c level and LA size in men, conflicts with current literature [32,33]. These two papers [32,33] are case-control studies including 40 and 60 patients with diabetes, respectively, and they are based on echocardiographic measurements, thus with a risk confirmation bias. The effect of diabetes on atrial size could be reduced when indexing LA size to body size, which is a major contributor to diabetes. A prior CT study including 3945 individuals, hereof 272 with diabetes, used LA size indexed to body size and found no association to diabetes in men, but an association to diabetes in women [29]. In our study, including 1166 men and 42 women with diabetes, we found a negative association between LA size and HbA1c among men, but not among women. In conclusion, the association between atrial size and diabetes may not be finally clarified. One other study has also reported a decreased LA size with increasing cholesterol level [29].

We found no rational physiological explanation for these surprising findings, but the results may be explained by inclusion of individuals from the general population in the DANCAVAS trial and our exclusion of individuals with known cardiovascular disease. Thus, we are studying healthy individuals without known cardiovascular disease, and it may be speculated that smoking, increased HbA1c and total cholesterol in the very beginning will induce fibrosis and shrinking of the LA. Over time, this will lead to contractile dysfunction and ultimate lead to dilatation and arrhythmias and cardiovascular disease. Thus, our findings could be partly explained by our exclusion criteria. However, the cross-sectional design does not permit causal inference.

The strength of our study is the large size of the study population comprising of randomly selected individuals with a high attendance rate; therefore, the risk of selections bias is low. Importantly, we excluded individuals with known cardiovascular disease, and our population consists primarily of Caucasian men aged 60–75 years; hence, the generalisation for females, other ethnic groups and different age groups can be questioned. An apparent limitation is the use of an area assessment rather than a full volume assessment of the LA. However, earlier studies have compared this axial measurement with 3D CT and MRI measurements of the left atrium and shown a high correlation [34,35]. Furthermore, CT measurements present high reproducibility as shown in our inter- and intra-observer agreement analysis whereas echocardiographic measurements are more difficult to reproduce [36]. This is also reflected by our analyses, as analyses based on the echocardiographic measurements were not as manifest as analyses based on CT measurements. LA size was measured at different stages of the heart cycle entailing a risk of underestimating LA size, if the scan was performed in the diastole. However, we found no substantial differences in subgroups of participants with a systolic versus a diastolic scan (Appendix A). Low R-squared values were seen in our results, which is expected in a model of healthy individuals where diseases known to influence LA size such as atrial fibrillation have been deliberately excluded. The participants were submitted to a somewhat stressful screening setting and to avoid over-diagnosing hypertension, the definition for hypertension was 160/100. This is not in accordance with the guidelines, but in our adjusted model, we did include pulse pressure instead of systolic or diastolic blood pressure. Excessive confounder adjustment was performed for all known risk factors, but the risk of residual confounding will always exist in such an observational study.

## 5. Conclusions

The associations between classic cardiovascular risk factor and LA size seen in this study may seem puzzling; however, they indicate that the pathophysiology behind atrial cardiomyopathy, remodelling, disease progression and consequently manifest cardiovascular disease may be even more complex than first assumed.

## Figures and Tables

**Figure 1 diagnostics-12-00244-f001:**
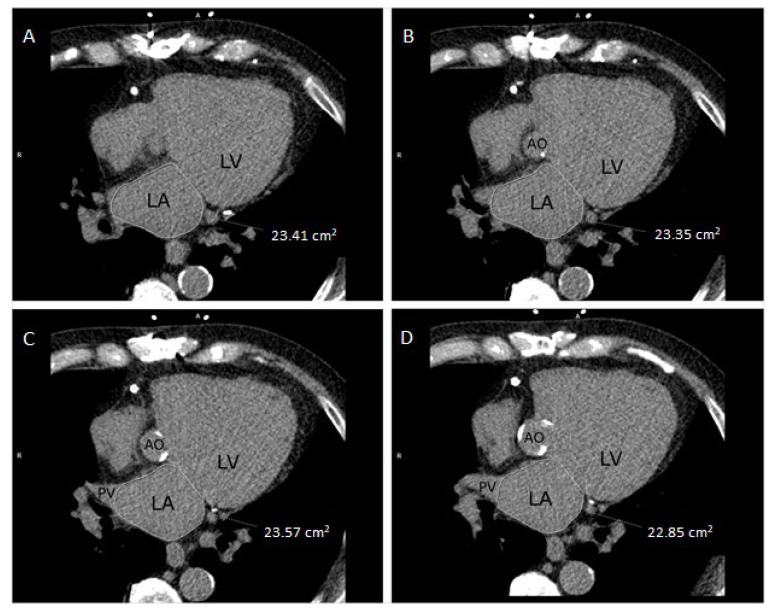
LA manually traced in multiple NCCT axial slices (**A**–**D**) to find the largest cross-section area (cm^2^). AO, aorta/aortic valve; LA, left atrium; LV, left ventricle; PV, pulmonary vein.

**Figure 2 diagnostics-12-00244-f002:**
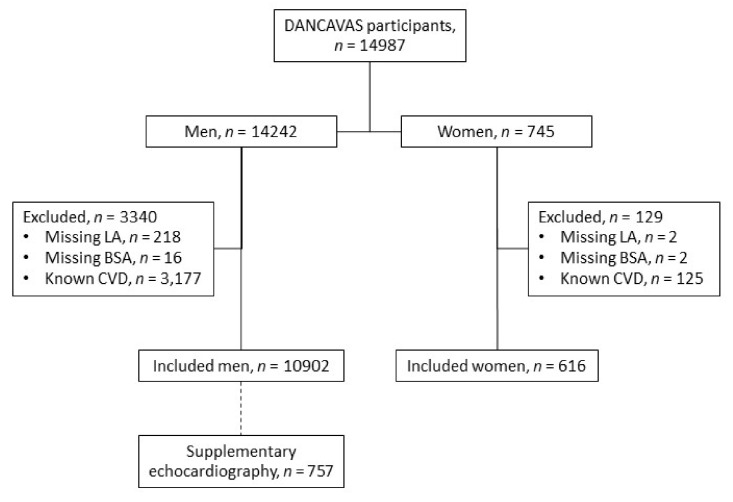
In- and exclusion criteria. BSA, body surface area; CVD, cardiovascular disease; LA, left atrium.

**Figure 3 diagnostics-12-00244-f003:**
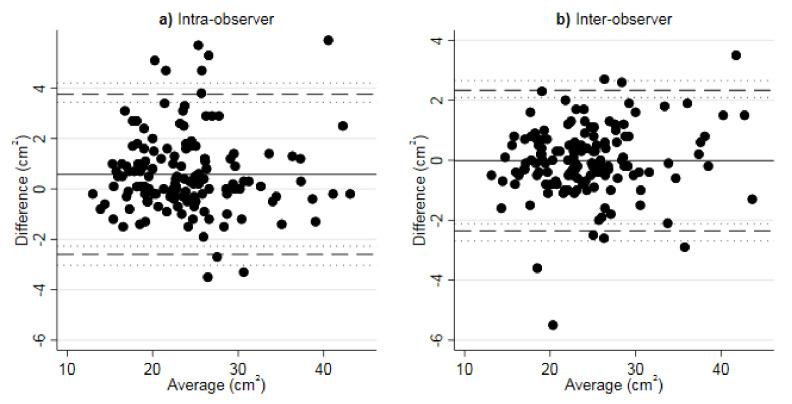
Bland–Altman plots comparing (**a**) intra-observer and (**b**) inter-observer agreement of LA area (cm^2^) measurements in NCCT (*n* = 140). Solid line: average difference (bias estimate). Dashed lines: 95% Bland–Altman limits of agreement. Dotted lines: exact 95% CI for Bland-Altman limits of agreement.

**Table 1 diagnostics-12-00244-t001:** Participant characteristic.

Variable	Men	Women	*p*-Value
*n*	10,902	616	
Age (years)	67 ± 4	68 ± 3	<0.001
LA size (cm^2^)	22.7 ± 5.1	19.4 ± 4.4	<0.001
LA index (cm^2^/m^2^)	11.1 ± 2.3	10.9 ± 2.3	0.19
BMI (kg/m^2^)	27.9 ± 4.2	26.5 ± 5.0	<0.001
BSA (m^2^)	2.0 ± 0.2	1.8 ± 0.2	<0.001
Smoker			
Active	1733 (16%)	76 (12%)	<0.001
Former	5376 (50%)	205 (33%)	
Never	3747 (35%)	334 (54%)	
Systolic blood pressure (mmHg)	150 ± 19	155 ± 20	<0.001
Diastolic blood pressure (mmHg)	83 ± 10	83 ± 9	0.38
Pulse pressure (mmHg)	67 ± 14	72 ± 16	<0.001
Hypertensive medication	4172 (38%)	253 (41%)	0.16
Thiazide	1166 (11%)	122 (20%)	<0.001
Beta-blocker	756 (7%)	68 (11%)	<0.001
ACE inhibitor/ARB	3134 (29%)	166 (27%)	0.34
Calcium antagonist	1821 (17%)	83 (14%)	0.04
Hypertension	6074 (56%)	385 (63%)	0.001
HbA1c (mmol/mol)	39 ± 7	39 ± 4	0.81
Diabetes mellitus	1166 (11%)	42 (7%)	0.002
Total cholesterol (mmol/L)	5.2 ± 1.0	5.8 ± 1.0	<0.001
LDL (mmol/L)	3.1 ± 0.9	3.2 ± 0.9	<0.001
HDL (mmol/L)	1.4 ± 0.4	1.7 ± 0.5	<0.001
Lipid lowering medication	2539 (23%)	190 (31%)	<0.001
Dyslipidaemia	8551 (78%)	555 (90%)	<0.001
eGFR (mL/min/1.73 m^2^)	79 ± 13	75 ± 13	<0.001
CAC score (AU)	85 (6–358)	9 (0–93)	<0.001
AVC score (AU)	3 (0–65)	0 (0–22)	<0.001
Ascending aortic diameter (mm)	37 ± 4	34 ± 4	<0.001

Data are expressed as mean ± SD or number (%). ACE, angiotensin converting enzyme; ARB, angiotensin II receptor blocker; AU, arbitrary unit; AVC, aortic valve calcification; BMI, body-mass index; BSA, body surface area; CAC, coronary artery calcium score; eGFR, estimated glomerular filtration rate; HDL, high density lipoprotein; LDL, low density lipoprotein; LA, left atrium.

**Table 2 diagnostics-12-00244-t002:** Linear regression of left atrium area index (cm^2^/m^2^) in men (*n* = 10,902).

	Univariate			Multivariable (R-Squared = 0.068)	
Variable	Δ LA Area Index (95% CI)	*p*-Value	R-Squared	Δ LA Area Index (95% CI)	*p*-Value
Age (years) *	0.08 (0.07; 0.09)	<0.001	0.016	0.06 (0.05; 0.07)	<0.001
Smoking					
Former	−0.40 (−0.50; −0.31)	<0.001	0.026	−0.42 (−0.51; −0.32)	<0.001
Current	−1.13 (−1.26; −1.00)	<0.001		−1.07 (−1.20; −0.94)	<0.001
Systolic blood pressure (mmHg) *	0.021 (0.019; 0.024)	<0.001	0.030	-	-
Diastolic blood pressure (mmHg) *	0.017 (0.012; 0.021)	<0.001	0.005	-	-
Pulse pressure (mmHg) *	0.029 (0.026; 0.032)	<0.001	0.031	0.026 (0.023; 0.029)	<0.001
Hypertensive medication	0.36 (0.27; 0.45)	<0.001	0.006	-	-
Hypertension	0.60 (0.52; 0.69)	<0.001	0.017	-	-
HbA1c (mmol/mol) *	−0.012 (−0.018; −0.006)	<0.001	0.0014	−0.016 (−0.022; −0.010)	<0.001
Diabetes mellitus	−0.16 (−0.30; −0.02)	0.02	0.0005	-	-
Total cholesterol (mmol/L) *	−0.08 (-0.13; −0.04)	<0.001	0.0014	−0.11 (-0.15; −0.07)	<0.001
LDL (mmol/L) *	−0.08 (−0.12; −0.03)	0.001	0.0010	-	-
HDL (mmol/L) *	0.14 (0.04; 0.24)	0.008	0.0006	-	-
Lipid lowering medication	0.004 (−0.010; 0.108)	0.93	0.0000	-	-
Dyslipidaemia	−0.14 (−0.24; −0.03)	0.01	0.0006	-	-
eGFR (mL/min/1.73 m^2^) *	−0.001 (−0.004; 0.002)	0.52	0.0000	-	-
CAC score (AU) *	0.0002 (0.0002; 0.0003)	<0.001	0.0047	-	-
AVC score (AU) *	0.0005 (0.0004; 0.0007)	<0.001	0.0046	-	-
Ascending aortic diameter (mm) *	0.11 (0.10; 0.13)	<0.001	0.0039	-	-

* By 1 unit increment. AU, arbitrary unit; AVC, aortic valve calcification; CAC, coronary artery calcium score; eGFR, estimated glomerular filtration rate; HDL, high density lipoprotein; LDL, low density lipoprotein.

## Data Availability

Data are stored at the Open Patient data Explorative Network (OPEN), https://open.rsyd.dk/redcap/, and is available upon request.

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
