# Peer review of "Association of Left Atrial Size Measured by Non-Contrast Computed Tomography with Cardiovascular Risk Factors—The Danish Cardiovascular Screening Trial (DANCAVAS)"

_diagnostics, 2022, doi:10.3390/diagnostics12020244_

Round 1
Reviewer 1 Report
The authors made the required changes. I suggest acceptance of the manuscript in the present form.
Reviewer 2 Report
The authors have adressed my comments. Minor spell checks remain.
This manuscript is a resubmission of an earlier submission. The following is a list of the peer review reports and author responses from that submission.
Round 1
Reviewer 1 Report
Fredgart et al. examined the association between indexed LA area measured by non-contrast CT and classical cardiovascular risk factors in participants without any overt cardiovascular disease. Age and hypertension were associated with LA size, but smoking, and also diabetes and dyslipidemia were associated with decreased LA size.
Individuals aged 60–75 years from the population-based multicenter Danish Cardiovascular Screening (DANCAVAS) trial were included in the study. The authors included since 2015 only men in the study. Even if they explain this due to unequal financial benefit for women, it remains unclear why this bias was necessary. However all (other) limitations oft he study are well described in the appropriate section.
All performed statistical methods in this study are valid and accurate.
The obtained findings have been consequently discussed, and compared to the published literature.
This article should be accepted for publication because it clearly describes its goals and delivers evidently and constructively the required information.
Reviewer 2 Report
The authors present a manuscript entitled “Association of Left Atrial Size Measured by non-contrast Computed Tomography with Cardiovascular Risk Factors – The Danish Cardiovascular Screening Trial (DANCAVAS)”. In this large-scale, population-based multi-center study, the authors investigated the correlation between indexed atrial area / size (derived from Calcium Scoring CT datasets) and cardiovascular risk factors. As major results, they found increasing indexed LA size with increasing age and pulse pressure, and decreasing indexed LA size associated with smoking, diabetes and total cholesterol.
Overall, populations-based imaging investigating cardiovascular risk factors and structural cardiac changes such as atrial size are relevant and scientific interesting. However, I have several methodological concerns, that need to be addressed before I think this publication can be acceptable for publications, including:
- There is a fundamental bias in LA area measurements by ECG gated Calcium Scoring Scans with various different protocols. Most scan protocols measured LA size in the ventricular diastolic phase with heart rates below 65/70 bpm (representing more the minimal atrial size), while LA size was measured in the ventricular systolic phase with heart rates over 65/70 bpm (representing the maximal atrial size). The authors combine those information and therefore present heavily biased data. It seems that the authors have evaluated subgroups for heart rates over and under 70 bpm (see limitations), but do not provide this relevant data. In my opinion, the analysis should be limited (or at least fully presented) for a homogenous measurement of atrial size (e.g. restrict the analysis to minimal or maximal atrial size). Second, as the cut-off for changing the ECG gating to ventricular systolic and diastolic gating is different in between scan protocols (varying in between non-existing, cut-off 65, cut-off 70 bpm), a simple thresh-holding of 70 bpm is most likely biased as well. Therefore, I would ask to present to split the cases to diastolic or systolic / “minimal / maximal” atrial size groups.
- Definition of hypertension is not reflective of guidelines and needs to be changed for the risk factor assessment.
- The authors should provide a subanalysis of atrial size derived by the echocardiographic data, as well as correlation in between CT and Echo assessment of atrial size
- Please conduct Kolmogorov Smirnov or Shapiro Wilk test for testing of normal distribution
- Please provide ICC as a measure of inter- and intra-observer agreement
- How would the results change when including patients with known CVD?
- The discussion should be re-assessed after performing the subgroup analysis mentioned above. In general, conflicting results should be discussed more in detail, especially in regard to results from prior investigations using CT or Echo. Please discuss e.g. technical differences etc.
- Language: Overall well written manuscript, some minor language revisions needed. Some typos, e.g. p9,l259 “HgA1c” etc.